# Correlation between Microvascular Damage and Internal Organ Involvement in Scleroderma: Focus on Lung Damage and Endothelial Dysfunction

**DOI:** 10.3390/diagnostics13010055

**Published:** 2022-12-25

**Authors:** Mario D’Oria, Ilaria Gandin, Pozzan Riccardo, Michael Hughes, Sandro Lepidi, Francesco Salton, Paola Confalonieri, Marco Confalonieri, Stefano Tavano, Barbara Ruaro

**Affiliations:** 1Department of Medical Surgical and Health Sciences, University Hospital of Cattinara, University of Trieste, 34149 Trieste, Italy; 2Biostatistics Unit, Department of Medical Sciences, University of Trieste, 34149 Trieste, Italy; 3Pulmonology Unit, Department of Medical Surgical and Health Sciences, University Hospital of Cattinara, University of Trieste, 34149 Trieste, Italy; 4Northern Care Alliance NHS Foundation Trust, Salford Care Organisation and Division of Musculoskeletal and Dermatological Sciences, Faculty of Biology, Medicine and Health, The University of Manchester, Manchester M6 8HD, UK

**Keywords:** systemic sclerosis (SSc), peripheral blood perfusion (PBP), dermal thickness (DT), lung impairments

## Abstract

Background. Systemic sclerosis (SSc) is an incurable connective tissue disease characterized by decreased peripheral blood perfusion due to microvascular damage and skin thickening/hardening. The microcirculation deficit is typically secondary to structural vessel damage, which can be assessed morphologically and functionally in a variety of ways, exploiting different technologies. Objective. This paper focuses on reviewing new studies regarding the correlation between microvascular damage, endothelial dysfunction, and internal organ involvement, particularly pulmonary changes in SSc. Methods. We critically reviewed the most recent literature on the correlation between blood perfusion and organ involvement. Results. Many papers have demonstrated the link between structural microcirculatory damage and pulmonary involvement; however, studies that have investigated correlations between microvascular functional impairment and internal organ damage are scarce. Overall, the literature supports the correlation between organ involvement and functional microcirculatory impairment in SSc patients. Conclusions. Morphological and functional techniques appear to be emerging biomarkers in SSc, but obviously need further investigation.

## 1. Introduction

Systemic sclerosis (SSc) is an autoimmune connective tissue disease, dominated by functional and structural abnormalities of the vessels and gradual fibrosis, targeting skin and internal organs [1,2,3]. The vascular abnormalities of the microcirculation are pivotal in the diagnostic process, the prognostic stratification, and the long-term monitoring of SSc patients. Microangiopathy is a gradual process that ends in damage of the vessels, deficient repair, and ischemia-reperfusion injury [1,2,4,5,6,7,8,9]. In SSc, the relations between morphological (i.e., microangiopathy) and functional damage (i.e., decrease in blood flow) has been proven by several studies [1,2,4,5,6,7,8,9].

Nailfold videocapillaroscopy (NVC) is a noninvasive, reliable, inexpensive, reproducible, and run-in method to define structural microvascular impairment in SSc [5,6,7,8,9,10,11]. However, standard NVC cannot measure capillary blood flow/perfusion, which can be quantified by many laser-based methods, thermography, and Doppler ultrasound, which is being evaluated as a potential means for assessing and quantifying peripheral blood perfusion. [6,7,8,9,10,11,12,13,14]. Furthermore, new technologies (i.e., optical Doppler tomography and spectroscopy) are emerging as promising tools to analyze peripheral skin blood flow [4,5,6,7,8,9,10]. Several studies have shown the relationship between NVC abnormalities and several specific clinical manifestations of SSc (e.g., digital ulcers-DU) and have identified NVC as a biomarker of treatment efficacy (e.g., for pulmonary arterial hypertension (PAH)) [14,15,16,17,18,19,20]. A few works have also demonstrated a correlation between gradual capillary depletion and the rate of SSc-related interstitial lung disease (ILD) [11,13,21,22,23,24,25,26,27,28,29,30,31]. The results of these studies are interesting because capillary density could be a potential predictor of developing major lung complications (i.e., SSc-ILD/PAH) [11,13,32,33,34,35,36,37,38,39,40,41,42,43,44,45]. However, a smaller number of studies have investigated the correlation between these SSc complications and blood perfusion as assessed by the functional techniques previously described (laser machines, thermography, and ultrasonography) [46,47,48,49,50,51,52,53,54,55,56].

This article aims to review the state-of-the-art literature on the techniques available to analyze microvascular abnormalities in SSc and its links to other clinical aspects of the disease. (i.e., ILD and PAH).

## 2. Nailfold Videocapillaroscopy (NVC)

Nailfold videocapillaroscopy (NVC) represents a non-invasive, validated, reproducible, and safe technique to define structural abnormalities occurring in the peripheral blood circulation in patients with SSc [1,2,4,5,6,7,8,9]. Given the early onset of structural and functional changes in the microvascular bed, which result in its typical clinical manifestation (Raynaud’s phenomenon—RP), the diagnostic, prognostic, and therapeutic implications of the structural assessment of the vessels provided by NVC allow for optimal clinical management. NVC identifies and quantifies microvascular abnormalities that differentiate secondary from primary RP and detects morphologic patterns characteristic of the different stages of SSc microangiopathy (“Early”, “Active”, and “Late”) [1,2,4,5,6,7,8,9] (Figure 1).

### 2.1. The Evaluation of Microvascular Damage by NVC

NVC is an effective tool to differentiate primary and secondary RP. Videapillaroscopy uses an optical microscope (magnification ~200×) and a digital video camera. The patient is placed in a sitting position in an acclimatized, temperature-controlled room (22–23 °C) for 15–20 min. The cuticles of the fingers are covered with a drop of immersion (cedar) oil (or equivalent), in order to reduce any refractive artifacts and to receive a better capillary visualization [1,2,4,5,6,7,8,9]. The morphological structure of the dermal nail capillaries is assessed by NVC, which is performed from the second to the fifth finger of both hands at the level of the periungual region (the thumb is excluded because of the nail thickness, which results in less transparency and the absence of a distal phalanx), as the capillaries are located just below the skin surface in this region, in a parallel manner [1,2,4,5,6,7,8,9]. NVC can reveal normal or pathological patterns of the capillaries in this region. In primary RP, the most common finding is a normal pattern; however, engorged efferent bundles or tortuous capillaries can be detected. In contrast, secondary RP is distinguished by structural abnormalities representing microvascular damage, such as avascular areas, giant capillaries, microhemorrhage, capillary leakage, and angiogenesis (Figure 1).

However, Cutolo et al. conducted a study of a large cohort, which showed that a subset (16%) of patients with an initial diagnosis of RP with normal NVC are prone to evolve into secondary RP over a mean follow up of 4.4 years [5]. In particular, patients with primary RP but with major nonspecific abnormalities detected at the first NVC evaluation (e.g., increased capillary diameter greater than 30 μm at arterial and/or venous branches, microhemorrhage, and reduced capillary number) should be monitored rigorously (e.g., every six months) because of the high risk of evolution into secondary RP. Nail fold capillaries have an average density of 9–12 capillaries per linear millimeter. In the early stage of SSc, they may be slightly reduced, while gradual desertification of nail capillaries represents a scenario of advanced SSc microangiopathy [5,7,14]. Physiologically, the capillaries of a healthy patient have the following characteristics: distribution parallel to each other, one (or two, more rarely) per dermal papilla, perpendicular to the nail fold. A normal capillaroscopic pattern presents a homogeneous distribution of hairpin-shaped capillaries, forming a “comb-like structure” [5,7,14]. The diameter of a normal capillary is usually <20 μm, whereas an ecstatic capillary has a diameter greater than 20 μm (but <50 μm) and a giant capillary is characterized by homogeneous dilatation of both afferent and efferent branches ≥50 μm. Recent work has shown that the presence of capillaries at arterial and/or venous branches with a diameter >30 μm is significant in subjects with primary RP who subsequently developed secondary RP, associated with SSc (mean follow up 3.56 years) [5,7,14,56]. Furthermore, the evidence at NVC of pathologic capillary dilatations in patients with primary RP should be considered a very early sign of evolution toward the an “Early” scleroderma pattern [56]. Another recent study by Cutolo et al. set out to examine nonspecific NVC abnormalities, particularly capillary dilatations, in patients with RP as potential precursors of the “scleroderma pattern” [14]. The authors observed that RP patients had significantly larger capillary diameters than healthy controls, both at baseline NVC and at follow up assessments (*p* = < 0.05 to <0.001).

The strongest single predictor of the evolution of the “scleroderma pattern” was the number of capillaries/mm with a diameter >30 µm at the basal NVC, with a progressive risk based on the number of capillaries involved (+24% per dilated capillary) (OR 1.24, 95% CI 1.17, 1.32). Furthermore, in this work, a tree-based analysis showed that the venous (efferent) diameter of the most dilated capillary in the basal NVC could be a potential variable of interest in identifying patients who are prone to maintain primary RP [14].

The term “scleroderma-pattern” encompasses a series of serial capillaroscopic changes that are distinctive of the microvascular involvement observed in >95% of patients with SSc [5,7,14]. NVC reveals the microangiopathic patterns characteristic of the “scleroderma pattern”. These specific capillaroscopic patterns have only been described in SSc [5,7,14].

### 2.2. The NVC Scleroderma Patterns

The patients with SSc, evaluated using NVC, have microvascular lesions that can be classified into three distinctive patterns that differ markedly from the normal morphological pattern and represent different phases of capillary alteration. They have a progressive progression from the first (“Early”), to the second (“Active”), and to the third (“Late”) microangiopathy capillaroscopic pattern [1,2,3,12,19,20] (Figure 1). Each stage has unique and typical morphological features and correlates with the autoantibody profile and disease duration [1,2,3,12,19,20].

The “Early” scleroderma pattern at NVC is the most significant, as it allows early diagnosis of SSc and scleroderma spectrum disease. The “Early” pattern is characterized by the following combination of features: few capillary microhemorrhages, few enlarged/large capillaries, an overall well-preserved capillary arrangement, and little or no capillary leakage (Figure 1). The “Active” one, on the other hand, has the following characteristics: common giant capillaries, frequent capillary microhemorrhages, slight capillary loss, modest architectural aberration of capillaries, and few or no branched capillaries (Figure 1).

Microhemorrhages are a common feature of both the “Early” and “Active” stages of SSc microangiopathy, as they are related to early vascular damage. In fact, they represent a transitional phase between the appearance of giant capillaries and the subsequent loss of capillaries. However, microhemorrhages are not pathognomonic of the scleroderma disease spectrum and, in fact, are a feature shared with other connective tissue diseases as well, such as systemic lupus erythematosus, myositis, and antiphospholipid syndrome and may have a post-traumatic genesis (e.g., antiphospholipid syndrome may have linear microhemorrhages parallel to the capillary line or manicure often causes misleading microhemorrhages) [1,2,3,12,19,20].

As for the “Late” scleroderma pattern at NVC, it is distinguished by the following features: irregular capillary engorgement, few or no giant capillaries and microhemorrhages, extreme capillary leakage with avascular areas, aberration of the standard capillary arrangement, and branched/accumulated capillaries [1,2,3,12,19,20] (Figure 1).

### 2.3. The NVC in Systemic Sclerosis: Clinical Application

A recent paper by Smith et al. reported that 257 of the 334 included SSc patients (76.9%) experienced a worsening of their clinical status, with the onset of severe new organ involvement or progression [13]. The authors reported that by using multivariable logistic regression, a normal capillary density at the NVC level was correlated with a better clinical outcome, particularly a lower rate of overall severe new organ involvement/progression [odds ratio (OR) = 0.77, *p* = 0.001] and less frequent peripheral vessel involvement (OR = 0.79, *p* = 0.043) [13]. In addition, microhemorrhages had an inverse relationship with the new onset of pulmonary hypertension (OR = 0.47, *p* = 0.029); however, a pattern of “severe” NVC (active/delayed) directly correlated with severe global new organ involvement/progression (OR = 2.14, *p* = 0.002) and skin progression (OR = 1.70, *p* = 0.049) [13].

There are examples in the recent literature regarding cross-sectional and predictive relationships between gradual capillaroscopic alterations/damage and clinical applications in SSc [19]. Several papers confirmed the predictive value of NVC for DU and other digital trophic lesions in SSc [57,58,59]. In particular, two prospective studies in two single centers have suggested potential prognostic indices, structured on the basis of quantitatively assessed distinctive scleroderma pattern parameters (e.g., loss of capillaries and/or giant capillaries), that should be able to predict impending progressive vasculopathy in patients with SSc [57,58,59]. Smith et al. also demonstrated the role of NVC in predicting DU and organ involvement in SSc [24].

The NVC analysis was established as a predictive tool for DU progression in SSc by an international “CAP” study on SSc (700 patients, 14 countries, and 59 centers) [60]. In addition, Ricceri et al. showed the correlation between capillaroscopic nail fold damage and the degree of pulmonary arterial hypertension [26,27]; 287 patients with SSc underwent NVC testing with the aim of recognizing a potential improvement in identifying patients at high risk for cardiopulmonary involvement. Accordingly, the NVC pattern was associated with ILD and cardiac/pulmonary involvement, independent of specific antibodies against extractable nuclear antigens [56].

Interestingly, NVC is applicable to follow the effects of treatments used in SSc. However, there are currently only a few pilot studies (nonrandomized, nondouble-blind, and nonplacebo-controlled) on this topic [15,61,62,63,64,65,66,67,68,69].

As a result of all these studies, NVC was added to the 2013 Early Diagnostic Criteria for SSc (VEDOSS) and the 2013 ACR/EULAR classification criteria for SSc, as well as the 2014 diagnostic criteria for primary RP [70,71,72].

The 2013 ACR/EULAR classification criteria for SSc encouraged the use of NVC in the routine clinical practice of physicians [71].

### 2.4. The NVC in Systemic Sclerosis: Clinical Application to Lung Impairment

The relationship between NVC changes and SSc-ILD detected using high resolution computed tomography (HRCT) has been discussed by several authors [28,29,73,74,75,76]. Forty-eight SSc-ILD patients with HRCT, ground-glass opacities, and/or fibrosis were the subject of a cross-sectional analysis by Caetano et al. [29]. The same group examined the relationship between NVC findings, the presence and severity of ILD, and functional decline of values at spirometry [28,29,73,74,75,76].

A substantial correlation was found between capillary loss and avascular regions and ILD. Worse values at spirometry were related to capillary depletion and avascular regions [28,29]. In another cross-sectional investigation, Guillen-del-Castillo et al. performed quantitative and qualitative analyses of 134 SSc patients (58 with ILD on HRCT) who had at least eight NVC scans (200 magnifications). In subjects with ILD SSc, a lower median capillary density (4.86/mm vs. 5.88/mm, *p*-value = 0.005) and higher median neoangiogenesis (0.56/mm vs. 0.31/mm, *p*-value = 0.005) were found. In addition, the patients with PAH showed higher neoangiogenesis (0.70/mm vs. 0.33/mm, *p*-value = 0.008). In a multivariate linear regression analysis, an association was shown between neoangiogenesis and decreased FVC (*p*-value = 0.001) and between the number of giant capillaries and decreased DLCO (*p*-value = 0.016). [28]. Guillen-del-Castillo et al. showed that the Late pattern was related to a decrease in FVC (*p*-value = 0.018) [29]. In a case–control study by Jehangir et al., 65 participants, including 10 subjects with primary Raynaud’s phenomenon (RP), 40 with SSc, and 15 controls matched for age and sex, were examined with a dermoscopy to examine the NVC pattern. Comparing the NVC patterns and HRCT images, only one participant with an Early NVC pattern had ILD, compared with 55% and 100%, respectively, of the Active and Late patterns [75].

In order to determine whether NVC could improve the identification of subjects at high risk of cardiopulmonary involvement, Markusse et al. performed it on 287 patients with SSc [18]. Of the research subjects, 51% had ILD, 59% had a reduced DLCO, and 16% had a pulmonary artery systolic pressure (sPAP) greater than 35 mmHg. The NVC pattern showed a consistent correlation with ILD or sPAP [18]. Decreased capillary density, which is related to a significant likelihood of developing PAH, and other NVC findings are indicators of SSc severity and development, respectively [30,31]. To better understand the relationship between pulmonary hemodynamic parameters and capillary density and diameter, Hofstee et al. examined 40 subjects with SSc (21/40 of them with SSc-PAH, according to right heart catheterization (RHC) evaluation), 20 idiopathic PAH participants, and 21 healthy subjects [33]. Although no statistically significant difference was found for capillary diameters (*p* value > 0.05), this analysis found that subjects with SSc-PAH had significantly lower capillary density than those without SSc-PAH (*p* value = 0.001). Similarly, Corrado et al. evaluated 39 participants with SSc (19/39 with SSc-PAH, identified by RHC as mPAP 25 mmHg, PCWP 15 mmHg, and pulmonary vascular resistance > 3 Wood’s units) and 25 healthy participants, 21 patients with idiopathic PAH, and 21 patients with this condition [34]. The researchers found that, in SSc subjects, the prevalence of PAH was significantly (*p*-value 0.05) inversely correlated with capillary density, directly correlated with both capillary diameter and giant capillaries (*p*-value 0.05), and correlated with aberrant capillary architecture (*p*-value 0.01) [34]. The RHC supported the results of the cross-sectional investigation by Riccieri et al., who examined NVC changes in 12 consecutive individuals with SSc-PAH. They showed that the degree of PAH was related to NVC deterioration. In fact, a statistically significant relationship was found between PAH, NVC score (which incorporates a semiquantitative score for density, size, bleeding prevalence, and architecture), and avascular area grading (*p*-value = 0.03 and *p*-value = 0.003, respectively) [26]. In addition, they noted that the participants with (compared with those without) SSc-PAH were more likely to have the active/late pattern (73% vs. 50%, *p*-value 0.05) [26]. The decreased capillary density and increased capillary width, two alterations observed by NVC, are strongly correlated with SSc-PAH, according to the two latest meta-analyses [30]. Kim et al. examined the link between clinical symptoms and quantitative computerized analysis of NVC. They found a significant relationship between capillary size/capillary loss and SSc-PAH (*p*-value 0.05) and DUs (*p*-value 0.01) [77]. In addition, a cross-sectional pilot study used a videocapillaroscope to assess sublingual microcirculation. The researchers found that healthy control participants, matched for age and sex, and PAH subjects had different indices of sublingual microcirculation flow and vascular tortuosity [78]. The prospective longitudinal investigation by Smith et al. examined 66 consecutive patients with SSc. They found a statistically significant correlation between NVC patterns and the possible occurrence of major pulmonary or microvascular involvement [36].

### 2.5. NVC and Peripheral Blood Flow

Although NVC cannot quantify blood perfusion in SSc under normal circumstances, laser-based approaches could be used to do so [36].

To contribute to the diagnosis, prognosis, and treatment of patients with SSc, NVC is used in conjunction with laser technology to provide a comprehensive morphological and functional assessment of the peripheral microvasculature. Because only a structural assessment of the vascular array can be performed, NVC cannot assess capillary blood flow/perfusion under typical conditions [36].

Consequently, some research has supplemented NVC with several additional methods (including laser techniques, thermography, and photoplethysmography) that can assess and quantify cutaneous blood flow [36].

Numerous studies have shown that individuals with SSc have reduced peripheral blood flow compared with healthy individuals and individuals with primary RP and that individuals with the “Late” NVC microangiopathy pattern have reduced blood flow at laser Doppler flowmetry (LDF) compared with individuals with “Active” and “Early” NVC [36].

## 3. An Overview of Different Functional Techniques to Evaluate Blood Perfusion

Laser-derived methods have increasingly gained ground in the perfusion and functional assessment of vascular abnormalities in SSc. The methods that exploit the Doppler effect, such as laser Doppler flowmetry (LDF) and laser Doppler imaging (LDI), allow for a noninvasive objective assessment of digital perfusion because of the ability to measure the Doppler shift induced by laser light scattering of red blood cells moving within skin vessels [52,79]. On the other hand, the methods that exploit the speckle pattern blurring effect, such as Laser Speckle Contrast Imaging (LSCI) and Laser Speckle Contrast Analysis (LASCA), measure the particle pattern shift created by laser light scattered by circulating red blood cells [79] (Table 1).

### 3.1. Laser Doppler Flowmetry

LDF uses a small probe, which operates only in adherence to the skin, limiting the field of evaluation to the probe site, resulting in high spatial variability, while providing excellent temporal resolution, being able to capture changes in perfusion in the real-time following provocation. Additionally, it lacks reproducibility [52,79].

### 3.2. Laser Doppler Imaging

LDI has good reproducibility and a wider field of view (down to the whole hand level) while having poor temporal resolution. The scanning laser can develop a perfusion map of the chosen visual field, but this is at the cost of a high scan time [52,79]. LDI can be used to determine the distal–dorsal difference, which is the perfusion gradient between the finger and the dorsum of the hand, calculated as the mean perfusion of the finger minus the mean perfusion of the dorsum. It follows that in patients with finger vascular lesions, such as those typical of SSc and particularly RP, this delta is negative [55].

### 3.3. Laser Speckle Contrast Imaging

LSCI is based on the two-dimensional quantification of speckle contrast changes of a divergent laser, allowing for full-field (up to the whole hand) and real-time assessment of tissue perfusion, overcoming the limitations of both LDF and LDI, emerging as a potentially superior method. Its penetration depth is about 300 μm (superficial papillary plexus of the skin), while it is deeper (about 1–1.5 mm) with laser Doppler techniques (deeper dermal blood flow). In addition, its blood perfusion results correlate with NVC results [52,79,80,81].

### 3.4. Laser Speckle Contrast Analysis

This technique is based, similar to LSCI, on the random inference pattern produced by backscattered laser light, called speckle pattern. However, it also implements a Charge-Coupled Device (CCD) camera, which can record and quantify inputs into technical parameters. In addition, it allows focusing on particular spatial (Region of Interest, ROI) and temporal (Time of Interest, TOI) frames.

LASCA can instantly assess the microvascular blood perfusion of tissues, thus the possible presence of functional damage, in a safe, non-contact, and non-invasive manner. Compared with LSCI, it has lower spatial resolution but higher temporal resolution, since in LSCI the contrast is calculated on a single pixel over a number of frames [53,82,83]. Compared with LDF, however, it is faster, more reproducible, and has higher spatial resolution [53]. One peculiarity of LASCA to note is that it is also a valid tool in black ethnic subjects [84].

### 3.5. Infrared Thermography (IRT)

This technique takes advantage of digital cameras to record and quantify the process of skin thermoregulation, effectively assessing digital perfusion, although it requires careful monitoring of acclimation time, camera–individual distance, ambient temperature, and humidity, as well as proper characterization of subjects (especially gender, smoking habit) [79,85,86,87,88]. Moreover, it can be combined with a local cold challenge: exposure to a localized cold stimulus largely mimics an RP attack, thus allowing dynamic vascular assessment [89]. The pathological findings of IRT have been shown to be predictive of an increased likelihood of developing DU and a more frequent need for surgical debridement [54,90], however, there is still no standardization [91].

### 3.6. The Emerging Role of UltraSound

In addition to laser techniques and thermography, the application of ultrasound to the diagnostic process and follow up of SSc-related small distal arteriole lesions is gaining ground. In fact, in addition to its use to assess skin thickness [55], evidence is emerging on this new purpose of ultrasound in SSc. Specifically, in patients with SSc, evaluation of the resistive index (RI) and peak systolic velocity (PSV) of interdigital arteries and nail arterioles using high-resolution ultrasound appears to be of some use in assessing subtle abnormalities in the downstream microvasculature [92]. However, pRI (Proximal Resistive Index) of the interdigital arteries does not seem to contribute to the detection of an ongoing pathological process in the small nail arterioles [92,93]. In addition, Power Doppler UltraSound (PDUS) may play a role in predicting digital ulcers by detecting ulnar artery occlusion and pathological blood flow in the finger pulp [21].

## 4. Correlation between NVC and Laser Techniques in the Examination of Peripheral Blood Flow

Murray et al. reported that NVC, LDI, and IRT are powerful and independent tools for discriminating between patients with SSc, patients with primary RP, and healthy controls, with NVC being the most effective classification method; however, when these three techniques are implemented together, the classification of SSc subspecies improves, with LDI and IRT being able to detect dynamic changes in skin microcirculation with comparable effectiveness [8]. Rosato et al., on the other hand, analyzed the synergistic effect of LDI and photoplethysmography (which uses infrared waves to detect changes in blood volume, with results positively concordant with those of LDI) in the assessment of skin perfusion of the hands and pulsatility of the digital arteries in both subjects with primary RP and SSc, reporting the superiority of the combined use of these techniques to distinguish these two different clinical entities. Regarding microvascular damage, the supremacy of NVC remains [94].

Moreover, a direct correlation with the degree of microangiopathy evidenced by NVC has been demonstrated for both LASCA and LDF [23,53]. Furthermore, patients classified as “Late” SSc, based on the pattern of microangiopathy revealed by NVC (Figure 1), when evaluated with both LASCA and LDF revealed a lower blood flow than subjects classified as “Active” or “Early”, further demonstrating the correlation between structural and functional damage [23,53]. Another study confirmed this correlation, using LASCA to assess blood perfusion: functional data at the level of the fingertips, periungual area, and palms were significantly lower in patients with SSc than in healthy subjects (Figure 2). In addition, perfusion values at the level of some skin areas revealed a clear correlation with the extent of nail microangiopathy in the same areas [23,53].

In addition, LASCA has been shown to be an effective and safe tool for monitoring the evolution of DU in SSc subjects by monitoring blood flow within the lesions and in the surrounding area, even during long-term follow up, finding a potential role in monitoring the efficacy of vasoactive therapies in re-establishing a proper proximal–distal gradient, which could then be related to the failure or clinical success of the therapeutic regimen [23,53]. Furthermore, since LASCA has already been shown to predict DU outcomes and effectively monitor ulcer evolution in SSc patients, it is conceivable to use the LASCA proximal–distal gradient (PDG) for monitoring the progression of vascular damage in SSc [95].

In their 2013 report, Della Rossa et al. employed LASCA on the dorsal surface of the hand to analyze qualitative factors, including the existence of a proximal–distal gradient and uniformity of flow distribution in 76 people (20 healthy subjects, 20 PRP, and 36 SSc patients) (Figure 3). When the participants were categorized according to the type of organ involvement, such as interstitial lung disease (ILD) or pulmonary arterial hypertension (PAH), no statistically significant discrepancies were observed in the baseline data, after the occlusive/ischemic test and after the cold test (*p*-value > 0.05) [96]. In 2016, Trombetta et al. obtained a similar result using LASCA to measure peripheral blood flow (PBP) in 30 SSc subjects undergoing long-term treatment with the endothelin receptor antagonist bosentan and the synthetic prostacyclin analog PGI2 iloprost. Interestingly, this drug regimen demonstrated a significant reduction in the rate of de novo DU and stability of both DLCO and PAH during follow up. However, in this case, a statistically significant link between PBP and DLCO or PAH was not found [51]. Gigante et al. (2021) evaluated PBP and hand PDG as biomarkers of major vascular complications (DU), PAH, and scleroderma renal crisis of SSc and mortality by LASCA [97]. Interestingly, researchers have observed that PDG can predict major vascular complications and 5-year mortality of SSc patients, but the predictive ability of LASCA of organ involvement in SSc patients was unclear, again [97].

In 2014, Sulli et al. first demonstrated a link between the degree of NVC, finger dermal thickness (ultrasound dermal thickness), and PBP by LDF [9]. The researchers found an inverse link between PBP and modified Rodnan skin score (mRSS) and ultrasound dermal thickness values (*p*-value = 0.007 and 0.0002, respectively). Compared with healthy subjects, SSc patients showed decreased PBP and increased ultrasound dermal thickness at the level of the fingers (*p*-value = 0.0001). Ruaro et al. (2018) evaluated 62 individuals with SSc and 62 healthy subjects using LASCA, ultrasound dermal thickness, and the modified Rodnan skin score (mRSS), which is a semiquantitative score used to assess skin thickness. The study showed a statistically significant inverse connection between skin PBP and ultrasound dermal thickness and mRSS at the finger level (*p*-value = 0.0005 and 0.0007, respectively) [2,45,98]. Before and after 10 days of local/systemic therapy, laser speckle contrast analysis (LASCA) was used to track DU perfusion in 20 SSc subjects with newly developed DU on the fingers [20]. Dedicated ROIs were drawn to examine PBP at the ulcer, peri-ulcer, periungual, and fingertip areas. During the follow up, ROIs defined at the level of the DU area (*p*-value 0.0001) and peri-ulcer area (*p*-value 0.0001) both showed a statistically significant increase in PBP. A statistically significant reduction in ulcer size was also observed in the LASCA assessment (*p*-value 0.0001) [20]. The effectiveness of LASCA in anticipating 40 DU outcomes in 31 patients with SSc was reviewed by Barsotti et al. in 2020. According to the LASCA evaluation, patients with dcSSc had significantly reduced mean blood flow rates on DU-affected fingers and around ulcers compared to patients with lcSSc (*p* values = 0.036 and 0.041, respectively) [95]. The blood flow on the finger with DU, at the ulcer, and in the peri-ulcer area was significantly higher in the presence of infection [95].

## 5. Flow-Mediated Dilation and Assessment of Endothelial Function

The term “endothelium-dependent flow-mediated dilation” (FMD) refers to the vasodilator reaction of a vessel to the increased shear stress associated with blood circulation. Nitric oxide (NO), one of several vasoactive compounds generated by endothelial tissue in reaction to shear stress, is of particular importance because a decrease in its bioavailability may contribute to the pathophysiology of vascular disease [99,100,101]. The FMD test, created in 1992, is currently the most widely used noninvasive method for assessing vascular endothelial activity in individuals.

The reactive hyperemia (RH) method, which causes vasodilation upstream of the conducted arteries in response to a brief increase in shear stress after the release of limb obstruction, has been the main method used to study the endothelial function of human conducted arteries by FMD (RH-FMD). In addition, FMD can be measured in reaction to prolonged increases in shear stress (otherwise known as prolonged stimulus, or SS-FMD), which are typically caused by limb warming or exertion. Exercise, in particular, generates a physiologically relevant stimulus, as daily activities increase shear stress and cause FMD. According to numerous studies, RH-FMD and SS-FMD are affected differently by different situations and acute treatments, with the latter occasionally presenting damage alone. According to the data, transient (RH) and sustained (SS) shear stress stimuli can be transduced through different signaling pathways. Accordingly, SS-FMD and RH-FMD seem to provide distinct perspectives on endothelial function [102]. Remarkably, with aging, the vasodilator response to reactive hyperaemia decreases, but it is unclear whether this is due to a change in the stimulus wall shear rate (WSR) or an altered response of FMD. Aizawa et al. demonstrated, using newer equipment that allows for the accurate assessment of WSR; elderly subjects exhibit a drastically altered and reduced WSR response to reactive hyperaemia compared with younger subjects [103]. Investigations into the relationship between FMD and microvascular abnormalities in SSc have also been conducted. Takahashi et al. used ultrasound to quantify brachial artery FMD and measure the percent increase in the brachial artery width after hyperemia in a study involving thirty-three SSc participants and twelve healthy controls. The FMD values in diffuse cutaneous SSc (dcSSc) patients (6.7 ± 4.0%) were equivalent to those found in limited cutaneous SSc (lcSSc) subjects and healthy controls but were substantially reduced in lcSSc subjects (5.3 ± 2.7 vs. 7.7 ± 2.0%, *p* < 0.05). However, a negative relationship was found between FMD values and disease duration in lcSSc patients (r = −0.64, *p* < 0.05), despite the fact that the FMD values did not correspond to any medical parameter in the dcSSc participants. In addition, the lcSSc subjects with lower FMD values had a higher prevalence of DU and higher right ventricular systolic pressure (for each, 75 vs. 10%, *p* < 0.05) than the others with regular FMD values. The researchers, therefore, concluded that the FMD values reflect the degree of vascular impairment in individuals with lcSSc, which progresses with the course of the disease and results in DU and PAH [104].

The sites most frequently affected by the RP phenomenon in SSc are the fingers and toes, the tip of the nose, and the ears, because they have unique anatomical and functional properties for regulating body temperature. While FMD is an indirect indicator of endothelial function, perfusion, and vasodilator capacity, digital thermal monitoring (DTM) of vascular reactivity assesses hyperemic, low-frequency radial artery blood flow, and finger vascular function. To improve the noninvasive assessment of endothelial function in SSc, Frech et al. examined the cross-sectional association between FMD and DTM factors in their research. FMD and DTM procedures were performed on 34 SSc participants on the same day. The shear rate (0.32, *p* = 0.07), relative FMD (0.42, *p* = 0.02), and absolute FMD (0.41, *p* = 0.02) showed weak but significant correlations with DTM. Reactive hyperemia had a slightly inverse but significantly correlated relationship with TMD (−0.44, *p* = 0.000). The circulation and basal diameter had no obvious relationship with DTM [105]. This noninvasive investigation of endothelial function in SSc reveals substantial correlations between macrocirculation (including relative and absolute FMD, shear rate, and peak hyperemia) and microcirculatory thermoregulation (represented by DTM), warranting further prospective investigation. The same authors previously evaluated comparing 52 age-matched healthy controls and 38 patients with SSc. Duplex ultrasonography was used to assess endothelial function, arterial anatomy, and peripheral hemodynamics. A blood pressure cuff was applied to the forearm and a 5 min ischemia was induced. The shear rate, endothelial function (FMD), and brachial artery vascular reactivity (peak hyperemia/area under the curve [AUC]) were assessed after occlusion. Compared with healthy volunteers, the SSc patients had narrower brachial arteries (*p* < 0.001), lower reactive hyperemia (*p* < 0.001), peak shear rate (*p* = 0.03), and brachial artery FMD (*p* < 0.001). Lower brachial artery FMD (*p* < 0.05) was observed in SSc patients with DU. The tertile analysis indicated that SSc subjects with the highest FMD tertile (>5.40%) had a 15% lower probability of manifesting DU than the two lowest FMD tertiles (5.40%), which had a 40–50% risk. Among the SSc patients with and without PAH, all the measures of brachial artery FMD were comparable (all *p* > 0.05) [106]. According to this study, the SSc patients have significantly narrower brachial arteries than the healthy controls, as well as reduced endothelium and peripheral vascular reactivity. Compared with subjects without DU, the SSc subjects with DU have significantly more severe deficits in endothelial function. FMD testing is therefore clinically useful in identifying SSc subjects at risk for DU. In the study by Roustit et al., 33 healthy controls, 36 patients with primary RP, and 42 patients with SSc were recruited. After a 5 min brachial artery occlusion, the brachial artery FMD and post-occlusive cutaneous reactive hyperemia (PORH) of each participant were evaluated simultaneously. The intima media thickness (IMT), pulse wave velocity (PWV), nitroglycerin-mediated dilation (NMD), and local thermal hyperemia were also measured. Compared with healthy volunteers, peak digital cutaneous PORH was abnormal in patients with primary RP and SSc, while FMD did not change substantially between groups. The peak digital cutaneous vascular conductance and brachial FMD were related in healthy subjects but not in patients with primary RP or SSc (r = 0.49; *p* = 0.004). Thermal hyperemia was impaired only in subjects with SSc. No differences in brachial NMD, IMT, or PWV were found between the groups. In summary, the subjects with SSc and primary RP showed a loss of the association between brachial FMD and digital cutaneous PORH, which might indicate that capillary function is impaired in SSc while the brachial artery endothelial function is intact [107].

Indeed, endothelial impairment and vasculopathy are both involved in the pathophysiology of SSc. In their investigation, Szucs et al. measured the intimal–medial thickness of the common carotid artery (ccIMT) in patients with SSc compared with healthy volunteers, as well as the flow-mediated endothelium-dependent dilation (FMD) and nitroglycerin-mediated endothelium-independent dilation (NMD) of the brachial artery. Using high-resolution ultrasound, the FMD and NMD of the brachial artery were assessed in 29 patients with SSc and 29 healthy controls. The FMD was substantially reduced (4.82 ± 3.76%) in the 29 patients with SSc (mean age: 51.8 years) compared with the control group (8.86 ± 3.56%) (*p* < 0.001). Among the patients (19.13 ± 17.68%) and controls (13.13 ± 10.40%), there was no difference in NMD (*p* > 0.1). Unlike the healthy people (0.57 ± 0.09), the SSc patients had a propensity to have a higher ccIMT (0.67 ± 0.26 mm), however, this difference was not statistically significant (*p* = 0.067). FMD and age were not correlated, but there was a significant and positive connection between ccIMT and age in the SSc (r = 0.470, *p* = 0.013) and in the healthy controls (r = 0.61, *p* = 0.003). In addition, ccIMT showed a significant link with disease duration (r = 0.472, *p* = 0.011) but not with FMD or NMD. NMD was significantly related to age in the SSc patients but not in the controls (r = −0.492, *p* = 0.012). Researchers have not detected the link between FMD, NMD, ccIMT, and SSc subtypes [108]. The reduction in FMD in SSc seems to be a sign of impaired endothelium-dependent vasodilation. Endothelium-independent dilatation detected using NMD is also still present, enabling the use of nitroglycerin treatment. Carotid atherosclerosis, indicated by ccIMT, may develop at a later age and after more prolonged disease. Consequently, the evaluation of FMD in the pre-atherosclerotic stage may be useful for diagnostic, prognostic, and therapeutic purposes. The patients with autoimmune diseases may actually have a higher vascular risk, which may increase mortality rates. Moreover, 101 patients with systemic autoimmune disorders (including primary antiphospholipid syndrome, systemic sclerosis, rheumatoid arthritis, and polymyositis), all with vasculopathies of various types, as well as 36 healthy people, participated in a study by Soltesz et al. In this study, flow-mediated brachial arterial vasodilation (FMD) and common carotid artery intima media thickness (ccIMT), which are markers of endothelial impairment and atherosclerosis, respectively, were compared with the augmentation index (AIx) and pulse wave velocity (PWV), which are markers of arterial stiffness. In contrast to control volunteers (FMD = 8.4 ± 4.0%; ccIMT = 0.6 ± 0.1 mm; Aix = −41.1 ± 22.5%; PWV = 8.0 ± 1.5 m/s; *p* < 0.05), autoimmune patients showed high resistance to aging (0.05), while autoimmune patients showed impaired FMD (3.7 ± 3.8%), elevated ccIMT (0.7 ± 0.2 mm), AIx (1.2 ± 32.2%), and PWV (9.7 ± 2.4 m/s). A strong inverse relationship was found between FMD and AIx (R = −0.64; *p* 0.0001) and PWV (R = −0.37; *p* = 0.00014). ccIMT and AIx (R = 0.34; *p* = 0.0009), ccIMT and PWV (R = 0.44; *p* < 0.0001) and AIx and PWV (R = 0.47; *p* < 0.0001) all showed significant positive relationships. The age of autoimmune patients was directly correlated with AIx, PWV, and ccIMT, while FMD was inversely correlated with age. In the subjects with autoimmune disorders, elevated AIx and PWV, a sign of arterial stiffness, may be highly correlated with endothelial impairment and overt atherosclerosis [109]. Further studies are needed to fully understand the complex assessment of vascular abnormalities in high-risk individuals using noninvasive techniques of measuring arterial stiffness, FMD, and ccIMT.

## 6. Conclusions

The morbidity and mortality rates of SSc are very high indeed. The usual hallmarks of SSc, namely microvascular damage and reduced peripheral blood flow, have important clinical implications both for disease progression and for the remarkable timeliness of diagnosis. The idea that microangiopathy may play a significant role in internal organ dysfunction, such as lung impairment, is supported by the data [10,20,28,29,73,74,75,76].

Recent years have seen a rapid advancement of new techniques to assess microvascular damage in SSc, which have the advantage of being noninvasive, reliable, reproducible, and measurable. Currently, the only standardized tool that can assess microvascular damage in SSc patients in routine practice is the NVC [10,20,28,29].

Several studies have found a strong relationship between specific quantitative and qualitative features of NVC, prevalence of ILD at HRCT, and pulmonary functional indicators such as FVC and DLCO. Thus, it is possible that microangiopathy plays a crucial role in the onset and development of SSc-ILD. Similarly, NVC can detect early microvascular abnormalities related to the presence of PAH and may be a key factor in the early diagnosis of SSc-PAH [10,20,28,29,82]. The relationship between vasculature, quantified by different approaches, and organ involvement, including lung damage, has also been documented in several investigations. Other complementary functional methodologies, which have already demonstrated their value in assessing the effect of drugs on skin blood perfusion in pilot clinical trials, still require validation. According to our review, the integration of morphologic and functional approaches allows for the accurate assessment of microvascular deterioration in SSc, as well as the ability to monitor disease progression and follow the response of the microvasculature to drugs. In addition, because of the poor prognosis of SSc patients with internal organ impairment, particularly those with respiratory symptoms, all such patients should be thoroughly evaluated from the early stage of the disease and followed up. To improve the diagnostic process, therapeutic strategy, and disease follow up in such a multifaceted disease, it is recommended that all physicians involved in the evaluation of patients with SSc use a combination of morpho-structural and functional methodologies [28,29,71,73,74,75,76,82,83]. However, further studies are needed to test each of these methods and verify their utility in the clinical setting. In conclusion, the body of evidence to date strongly implies that NVC may be a potential biomarker in SSc, which obviously deserves further investigation.

## Figures and Tables

**Figure 1 diagnostics-13-00055-f001:**
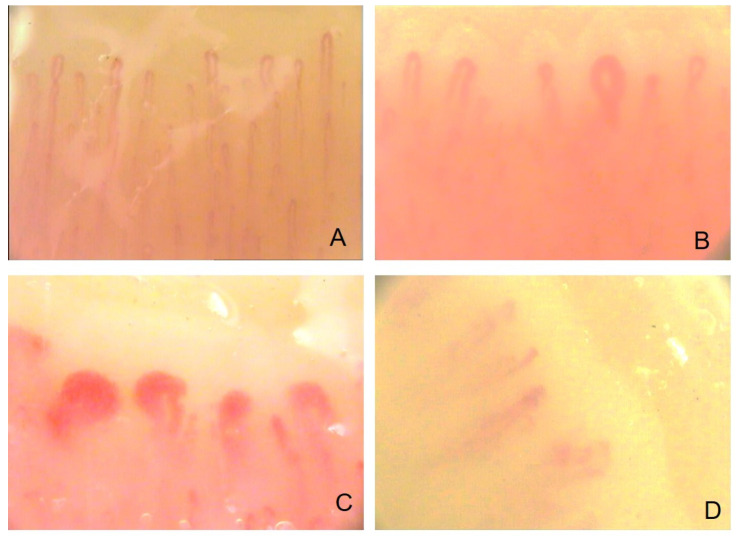
Nailfold videocapillaroscopy images (×200) in healthy subjects (**A**), “Early” (**B**), “Active” (**C**), and “Late” (**D**) patterns of scleroderma microangiopathy.

**Figure 2 diagnostics-13-00055-f002:**
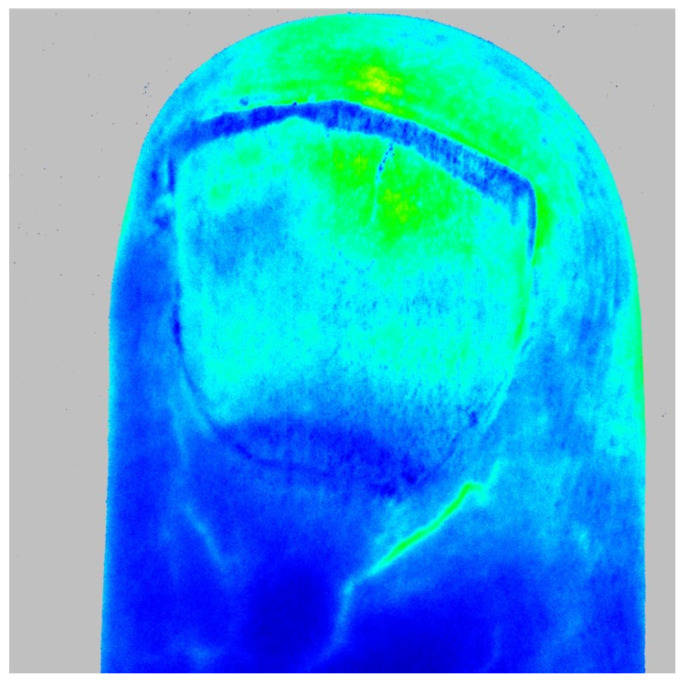
Laser speckle contrast analysis images of the distal–dorsal aspect of a finger in a patient with a “Late” scleroderma pattern of microangiopathy. Blue = low blood perfusion, yellow = intermediate blood perfusion, and red = high blood perfusion.

**Figure 3 diagnostics-13-00055-f003:**
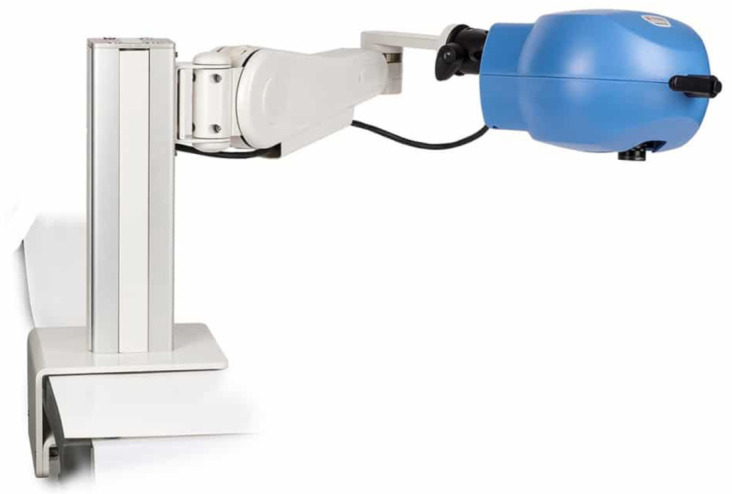
Laser speckle contrast analysis (LASCA) is composed of: a head, an adjustable arm, a detector camera, and a software for data acquisition and analysis. (Courtesy by Perimed).

**Table 1 diagnostics-13-00055-t001:** Main characteristics of available techniques to evaluate morphological and functional microvascular damage.

Technique	Principle	Skills	Disadvantages	Linked Organ Damage
Laser speckle contrast analysis (LASCA) and imaging (LSCI)	When the laser light illuminates the tissue, the static areas result in a stationary speckled pattern, which fluctuates due to the movement of the red blood cells, causing them to appear blurred and thus creating an overall speckled pattern. Based on the level of blurring (contrast), the degree of blood perfusion is determined (low contrast = high perfusion).	-Fast data acquisition -Good spatial resolution (images rich in detail) -Possibility of both spatial (areas) and temporal (dynamic response) evaluations -Non-contact technique -Valid instrument in subjects of Black ethnicity -Decreased PBP at the fingertips, periungual, palmar aspect of third finger and palm areas in SSc patients -After cold test, SSc patients had a significant reduction in blood flow and a higher recovery time compared to HS and PRP -Increase in PBP during long-term therapy with the endothelin receptor antagonist bosentan (BOSE) and the synthetic analog of prostacyclin PGI2 iloprost (ILO) in a 4-years follow up in patients with Sac	Measures perfusion at tissue and skin structure levels (capillaries, arterioles, venules, and shunts).	Microvascular damage in SSc, PAH (assessment of pulmonary damage)
Laser Doppler Flowmetry (LDF)	Assesses skin perfusion by measuring the Doppler effect induced by coherent light scattering related to red blood cell movement.	-Heatable fiber optic probe -Can detect both tissue perfusion and skin temperature -Optimal time resolution	-Measurement of blood flow at one point -Lack of reproducibility -Contact technique, limited to probe site -Measurement of local tissue perfusion (capillaries, arterioles, venules, shunts)	Microvascular damage in SSc
Laser Doppler Perfusion Imaging (LDPI)	-A laser with a standard output wavelength (color) directed against a diffusion medium such as skin or blood -A Doppler shift (color change) is induced by moving objects, such as blood cells, in scattered light	-Measurement of blood flow of an entire area -Non-contact technique -Good reproducibility	-Slow data acquisition -Measurement of skin perfusion (capillaries, arterioles, venules, and shunts) -Poor temporal resolution	PRP and vascular involvement in SSc
Thermography	Indirect measurement of blood flow	Non-contact technique	Indirect measurement of blood flow	Microvascular damage in SSc
Near infrared spectroscopy 2D imaging (NIRS-2D)	-Assess regional tissue oxygenation and microvascular function, in conjunction with vascular occlusion testing	-Non-invasive technique -Evaluation of vascular function by measuring blood flow	-Indirect measurement of blood flow -Low spatial resolution and poor sensitivity	Microvascular damage in SSc
Photoplethysmography (PPG)	It uses infrared light to measure changes in blood volume	Non-invasive technique	Its application in Raynaud’s phenomenon to evaluate digital artery response to cold test and medications has so far been limited	Microvascular damage in SSc
Infrared thermography (IRT)	Through digital thermal cameras, it records and quantifies the skin thermoregulation process, evaluating digital perfusion.	Predictor of higher probability of developing DU and need of surgical debridement	It requires careful monitoring of acclimatization time, camera-to-individual distance, ambient temperature, and humidity, as well as proper subject characterization.	Evaluation of PRP in patients with SSc
Nailfold videocapillaroscopy (NVC)	Method for an early diagnosis and follow up of nailfold microangiopathy combining a microscope (with a 5-million-pixels system) and a digital video camera.	Noninvasive, safe, inexpensive, reproducible, and validated method to assess morphological impairment in SSc	Need for further investigation	Microvascular damage in SSc and efficacy of treatment, PAH, ILD (assessment of pulmonary damage and heart involvement)

HS, healthy subjects; PBP, peripheral blood perfusion; SSc, systemic sclerosis; PRP, primary Raynaud’s phenomenon; IRT, infrared thermography; SLE, systemic lupus erythematosus; and DU, digital ulcers.

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
