# Peer review of "Correlation between Microvascular Damage and Internal Organ Involvement in Scleroderma: Focus on Lung Damage and Endothelial Dysfunction"

_diagnostics, 2022, doi:10.3390/diagnostics13010055_

Round 1

Reviewer 1 Report

Very interesting and comprehensive manuscript. The discussion was well structured and articulated. Improving English language (minor corrections). Add references to the conclusion section. 

Author Response

We would like to thank the reviewer for the comments. The manuscript has been improved regarding the English language (minor corrections), as requested. We have been added references to the conclusion section, as requested. All suggested and implemented changes have been underlined in the text in red.

Reviewer 2 Report

The article represents an interesting presentation of new diagnostic possibilities in monitoring functional abnormalities in microcirculation in SSc. At the same time, the importance of the association of these abnormalities with lung changes, digital ulcerations and endothelial dysfunctionwas pointed out, as well as the association of these microcirculation abnormalities with the status of antibodies, which are the most common in SSc. There are a lot of details in the paper that are very difficult to follow. I suggest that certain parts need to be shortened, given a clearer form for easier monitoring and understanding ( NVC in systemic sclerosis: clinical application to lung impairment; Correlation between NVC and laser techniques in the examination of peripheral blood flow, Flow-mediated dilation and assessment of endothelial function) Syntactic and semantic correction are needed, as well as language correction.

A detailed proposal for textual corrections is given in the attachment (word format)

Author Response

We would like to thank the reviewer for all comments.
In accordance with all suggestions, some parts have been shortened,
giving a clearer form for easier monitoring and understanding
(see NVC in Systemic Sclerosis: Clinical Application to Pulmonary Compromise;
Correlation of NVC and Laser Techniques in Peripheral Blood Flow Examination ,
flow-mediated dilatation and assessment of endothelial function)
In addition, syntactic and semantic corrections have been made,
as well as an extensive revision of the English language and style by a native speaker. All suggested and implemented changes have been underlined in the text in red 

Round 2

Reviewer 2 Report

I appreciate the corrections

The author should read the text carefully and correct letters mistakes,  which still exist in the text

Interesting and comprehensive work